# Insights into Non-Proteolytic Inhibitory Mechanisms of Polymorphic Early-Stage Amyloid β Oligomers by Insulin Degrading Enzyme

**DOI:** 10.3390/biom12121886

**Published:** 2022-12-16

**Authors:** Karina Abramov-Harpaz, Yifat Miller

**Affiliations:** 1Department of Chemistry, Ben-Gurion University of the Negev, P.O. Box 653, Beer-Sheva 84105, Israel; 2The School of Brain Sciences and Cognition, Ben-Gurion University of the Negev, Beer-Sheva 84105, Israel; 3Ilse Katz Institute for Nanoscale Science and Technology, Ben-Gurion University of the Negev, Beer-Sheva 84105, Israel

**Keywords:** molecular dynamics simulations, amyloids, Alzheimer, insulin-degrading enzyme

## Abstract

Insulin degrading enzyme (IDE) has been detected in the cerebrospinal fluid media and plays a role in encapsulating and degrading the amyloid β (Aβ) monomer, thus regulating the levels of Aβ monomers. The current work illustrates a first study by which IDE encapsulates polymorphic early-stage Aβ oligomers. The main goal of this study was to investigate the molecular mechanisms of IDE activity on the encapsulated early-stage Aβ dimers: fibril-like and random coil/α-helix dimers. Our work led to several findings. First, when the fibril-like Aβ dimer interacts with IDE-C domain, IDE does not impede the contact between the monomers, but plays a role as a ‘dead-end’ chaperone protein. Second, when the fibril-like Aβ dimer interacts with the IDE-N domain, IDE successfully impedes the contacts between monomers. Third, the inhibitory activity of IDE on random coil/α-helix dimers depends on the stability of the dimer. IDE could impede the contacts between monomers in relatively unstable random coil/α-helix dimers, but gets hard to impede in stable dimers. However, IDE encapsulates stable dimers and could serve as a ‘dead-end’ chaperone. Our results examine the molecular interactions between IDE and the dimers, and between the monomers within the dimers. Hence, this study provides insights into the inhibition mechanisms of the primary nucleation of Aβ aggregation and the basic knowledge for rational design to inhibit Aβ aggregation.

## 1. Introduction

Alzheimer’s disease (AD) is a neurodegenerative disease characterized by the accumulation of misfolded 40–42-residue amyloid-β (Aβ) into insoluble neurotoxic fibrillar aggregates with a rich β-sheet structure. Since AD is the most common neurodegenerative disorder and affects a major part of the world’s population [1], there is a great interest to inhibit or prevent Aβ aggregation. One of the possible suggestions of inhibiting Aβ aggregation focuses on molecular chaperones and heat-shock proteins (HSPs), because Aβ has various Aβ-peptidases [2,3]. Insulin-degrading enzyme (IDE) is a Zn^2+^-metalloprotease that is ubiquitously secreted in human tissues and organs, including in the brain. IDE presents at high levels mainly intracellularly in the cytosol, thus suggesting that it has diverse functions [4,5]. It has been suggested that IDE operates as HSPs [6]. Moreover, extensive studies have indicated that IDE is associated with AD [7,8,9,10,11,12]. Both clinical and in vivo studies have demonstrated that low levels of IDE in the brain of AD patients induce the progression of AD [7,8,9]. Thus, it is suggested that IDE may be a therapeutic strategy to treat AD patients [10,11,12]. Furthermore, in vivo and in vitro studies illustrate that IDE degrades monomers of amyloidogenic peptides such as insulin, Aβ, and amylin [13,14,15,16]. Thus, the degradation of Aβ, for example, may notify that IDE inhibits Aβ aggregation. Moreover, it has been shown that cross-seeding between Aβ and amylin or insulin may explain the link between type 2 diabetes (T2D) and AD [17,18,19,20,21,22,23]. The link between these two diseases may also be suggested by the fact that IDE degrades Aβ, which is related to AD, and amylin and insulin, which are related to T2D.

IDE is characterized by four homologous domains: domain 1 (residues 43–285), domain 2 (residues 286–515), domain 3 (residues 542–768), and domain 4 (residues 769–1016). The N-terminal domains (1 and 2) form a αβαβα sandwich (IDE-N), and the C-terminal consists of domains 3 and 4 (IDE-C). The IDE-N and the IDE-C are connected by a 28-residue loop. The IDE comprises an internal chamber between the IDE-N and the IDE-C domains, that allows binding of the substrates and degrading them. Two main conformations of IDE were identified: open-state and closed-state. The open-state allows for the diffusion of substrates in and out of the chamber, while the closed-state either entraps the substrates that entered or prevents the access of substrates to the chamber [24].

In vitro and in vivo studies have demonstrated that IDE forms a highly stable complex with the Aβ monomer [25]. It has been suggested that the entrapment of substrates in IDE chamber is involved with intermolecular interactions with IDE [25]. Recently, it has been proposed that residues 10–19 of Aβ bind to IDE [3]. However, the intermolecular interactions between IDE and the Aβ monomer at the atomic level has not been investigated to date. The entrapment of the Aβ monomer in the IDE chamber assists in the clearance of Aβ from cells, thus decreasing the fibrillation alongside the “dead-end chaperon” function of IDE [25]. The entrapment of the Aβ monomer has been also investigated in the chamber of the mutant IDE (E111Q) [26]. This inactive IDE mutant generates nonfibrillar Aβ aggregates that are not toxic through amyloidogenic pathway blocking, consequently inhibiting amyloid aggregation. Finally, a recent study has investigated the proteolytic activity of Aβ by both wild-type IDE and the mutant cysteine-free E111Q-IDE in the presence and absence of zinc ions [27]. In the absence of zinc ions, this mutant degrades Aβ monomers and creates non-fibrillar Aβ aggregates that are not toxic [27]. Hence, the formation of the non-toxic aggregates occurs by entrapment of Aβ monomers by wild-type IDE or mutants of IDE.

While Aβ monomers play a role in important neuroprotective functions in the brain [28,29], the early-stage oligomers are known as toxic species that yield to the pathology of AD. To date, the entrapment of Aβ oligomers by wild-type IDE or mutants of IDE has not been investigated by experiment nor by molecular modeling tools. In vitro study has shown that IDE can entrap α-synuclein (AS) oligomers and inhibit (AS) fibrillation [30]. Thus, there is a tremendous interest in inhibiting the formation of early-stage Aβ oligomers by IDE entrapment and by inhibiting the contacts between Aβ monomers that play a role in primary nucleation.

The major aim of the current study was to explore the mechanistic effect of IDE on the stability of Aβ oligomers. Herein, we report the first study that shows the ability of wild-type IDE to entrap polymorphic Aβ early-stage dimers. Moreover, the effect of wild-type IDE on the inhibition of the polymorphic Aβ dimers has been explored at the molecular level. Finally, the molecular mechanisms of the inhibitory activity of the polymorphic Aβ dimerization by the non-proteolytic action of IDE were first proposed.

## 2. Methods and Materials

### 2.1. Construction of IDE Model

The modeling procedure was performed by using the Biovia Discovery Studio Visualizer 3.5 version [31]. The dimeric form of IDE that is complexed with insulin was solved by cryo-EM (PDB ID code: 6BF8) [32]. This dimeric form consists of two IDE conformations: open-state and close-state. The open-state conformation IDE was used in the current work, because it has a relatively large chamber to allow insertion of the Aβ dimer into the chamber. This Cryo-EM IDE structure lacks the Zn^2+^ ion in the catalytic zinc-containing binding site. Thus, the Zn^2+^ ion was bound to the catalytic zinc-containing binding site: residues H108, H112, E189. In our previous study, we simulated this IDE structure and determined a new conformation of IDE [33].

### 2.2. The Rational of the Choice of Aβ Dimer Models

It is well-known that Aβ fibrils are polymorphic [34], thus it is expected that Aβ oligomers are also polymorphic. Moreover, conformational change also occurs in a particular Aβ polymorph oligomer. Hence, it is very challenging to investigate a wide range of oligomers due to the computational costs. To address the limitation of the computational costs, it is crucial to choose feasible structural polymorphic oligomers and to justify the choice. Since the current work was focused on Aβ dimers, one needs to choose feasible dimers from a study that illustrates polymorphic Aβ dimers.

Previously, our group revealed polymorphic Aβ dimers [35], in which two types of Aβ dimers were investigated: random coil/α-helix dimers and fibril-like dimers. Two random coil/α-helix dimers were examined: model A1—the monomers were arranged in a parallel orientation, and model A2—the monomers were arranged in an antiparallel orientation. Simulations of model A1 generated five distinct dimers (models C1–C5), while model A2 accomplished four distinct dimers (models D1–D4). These four dimers D1–D4 were more greatly populated than the five dimers C1–C5. Models D1 and D4 demonstrate relatively large structural differences among these four models. Hence, these two models were applied in the current work to examine the effect of IDE on these two random coil/α-helix dimers.

In addition, two Aβ fibril-like dimers were explored [35]: in model A3, the monomers were arranged in a parallel orientation, while in model A4, the monomers were arranged in an antiparallel orientation. Model A3 was relatively more populated than model A4. Thus, the most populated Aβ dimer (model A3) was applied in the current work to examine the effect of IDE on the Aβ fibril-like dimer.

### 2.3. Construction of IDE-Aβ Fibril-Like Dimer Complex Models

The fibril-like Aβ dimer-model A3 was inserted into the chamber of the IDE, while avoiding clashes between atoms of the IDE and Aβ dimer. To illustrate the domains of Aβ dimers that are inserted into the IDE chamber, we divided Aβ into two domains: (i) the N-terminal domain: residues D1–K16, and (ii) the C-terminal domain: residues L17–A42. Three models of IDE-Aβ fibril-like dimer complexes were constructed. In model F1, the C-termini domains of the fibril-like dimer were mostly contacted to the IDE-C domains (Appendix A), while in model F2, the C-termini domains of the fibril-like dimer mainly interacted to the IDE-N domains (Appendix A). Finally, in model F3, the C-termini domains of the fibril-like dimer were exposed to the solution, and the N-termini domains of the monomers were inserted to the IDE chamber, producing interactions mainly with domains 1, 2 and 3 (Appendix A).

### 2.4. Construction of IDE-Aβ Random Coil/α-Helix Dimer Complex Models

The random coil/α-helix dimers, D1 and D4, structurally do not form a well-organized cross-β structure. Moreover, these dimers are more bulk and less packed, therefore it is challenging to insert these dimers into the IDE chamber. The insertion of these dimers led to initial contacts of the dimers with several parts of the IDE (Appendix A). A total of four models of IDE-Aβ random coil/α-helix dimer complex were constructed. In models R1 and R2, the random coil/α-helix Aβ dimer D1 was inserted into the IDE chamber. In models R3 and R4, the random coil/α-helix Aβ dimer D4 was inserted into the IDE chamber.

In model R1, one monomer binds to domain 4 of IDE and the second monomer binds mainly to domain 1 of IDE. In model R2, one monomer binds to IDE-N (domains 1 and 2), and the second monomer mainly binds to domain 4. In model R3, one monomer binds to domains 2 and 3, and the second monomer binds to domains 1 and 4. Finally, in model R4, one monomer binds to domain 4, and the second monomer binds mainly to domains 2 and 3.

### 2.5. Molecular Dynamics (MD) Simulations Protocol

The MD simulations of the solvated constructed models were performed in the NPT ensemble using the NAMD package [36], with the CHARMM36 force-field with the CMAP correlation [36]. The IDE-Aβ dimer complex structural models were energy minimized and explicitly solvated in a TIP3P water box [37,38]. Each water molecule within 2.5 Å of the models was removed. Counter ions (NaCl) were added at random locations to neutralize the charge of the models. The Langevin piston method [36,39,40] with a decay period of 100 fs and a damping time of 50 fs was used to maintain a constant pressure of 1 atm. The temperature 330 K was controlled by a Langevin thermostat with a damping coefficient of 10 ps [36]. The short-range van der Waals (VDW) interactions were calculated using the switching function, with a twin range cutoff of 10.0 and 12.0 Å. Long-range electrostatic interactions were calculated using the particle mesh Ewald method with a cutoff of 12.0 Å [41,42]. The equations of motion were integrated using a leapfrog integrator with a step of 1 fs. The counter ions and water molecules were allowed to move. The hydrogen atoms were constrained to the equilibrium bond using the SHAKE algorithm [43]. The minimized solvated systems were energy minimized for 5000 additional conjugate gradient steps and 20,000 heating steps at 250 K, with all atoms allowed to move. Then, the system was heated from 250 K to 300 K and then to 330 K for 300 ps and equilibrated at 330 K for 300 ps. The choice of a higher temperature than physiological temperature was to investigate the stability of the constructed models. Obviously, structural models that are stable at higher temperature will also be stable at the physiological temperature. Simulations ran for 200 ns for each variant model. The structures were saved every 10 ps for analysis.

### 2.6. Structural Analyses

The structural stabilities of the models were explored by using several analyses. The convergence of Aβ monomers within the dimer and the four domains within IDE were analyzed by the root-mean-square-deviation (RMSD) analysis. To evaluate the fluctuations of each residue within IDE and Aβ, root-mean-square-fluctuation (RMSF) analysis was performed. To compare the secondary structure (of both Aβ dimers and IDE domains) between the initial IDE-Aβ dimer complex and the final simulated complex, the database of the secondary structure of protein (DSSP) method [44] was applied. The values of the DSSP for the initial complex were taken from the first 5 ns of the simulations, while the values of DSSP for the final simulated complex were taken from the last 5 ns of the simulations. The DSSP method provides the percentage of the α-helix or β-strand that is located along the sequence of the Aβ monomer/dimer and IDE. To provide insights into the dynamics of the IDE along the MD simulations, distances between residues within the IDE-N domains and IDE-C domains were measured.

The inter- and intra-peptide interactions were measured by the distance between atoms of residues along the MD simulations. For hydrophobic interactions, the distance was measured between the Cα atoms of two hydrophobic residues. The cutoff distance for these interactions was 10 Å [45]. The hydrophobic contact maps between Aβ monomers are described by the occurrence of the interactions along the MD simulation. The contacts are illustrated by three levels: strong, weak, and none. These levels were determined by four ‘rules’: (1) the orientation of the side chain of the hydrophobic facing the relevant hydrophobic side chain residue; (2) the distance cutoff between Cα atoms was up to 10 Å; (3) number of contact interactions between the relevant hydrophobic residues; and (4) the occurrence percentage along the MD simulations. Hence, in cases where the residues faced each other, the number of the interactions was ≥2 and the percentage occurrence of these interactions along the MD simulations was ≥85%, the contacts were considered as a high level. In cases where the residues faced each other, the number of the interactions was ≤2 and the percentage occurrence of these interactions along the MD simulations was ≤85%, the contacts were considered as a weak contact. In cases where none of these rules prevailed, there were no contacts. Finally, electrostatic interactions and hydrogen bonds between residues within the IDE and Aβ monomers were measured along the MD simulations. The cutoff distances for the electrostatic interactions were set to 4 Å [46] and the hydrogen bonds were set to 2.4 Å [47].

### 2.7. Determining the Conformational Energies and Populations for the Simulated IDE-Aβ Dimers

To obtain the relative conformational energies of the Aβ various conformations, the dimer trajectories of the last 5 ns were first extracted from the explicit MD simulation excluding the water molecules. The solvation energies of all systems were calculated using the generalized born method with molecular volume (GBMV) [48,49]. The hydrophobic solvent-accessible surface area (SASA) term factor was set to 0.00592 kcal/mol∙Å^2^. Each variant was minimized to 1000 cycles and the conformation energy was evaluated by grid based GBMV. The minimization did not change the conformations of each variant, but only relaxed the local geometries due to thermal fluctuation, which occurred during the MD simulations. A total of 3500 conformations (500 conformations for each of the seven examined conformers) were used to construct the free energy landscape of the conformers and to evaluate the conformer probabilities by using Monte Carlo (MC) simulations. In the first step, one conformation of conformer i and one conformation of conformer j were randomly selected. Then, the Boltzmann factor was computed as e^−(E_j_ − E_i_)/kT^, where E_i_ and E_j_ are the conformational energies evaluated using the GBMV calculations for conformations i and j, respectively, k is the Boltzmann constant, and T is the absolute temperature (298 K used here). If the value of the Boltzmann factor is larger than the random number, then the move from conformation i to conformation j is allowed. After one million steps, the conformations ‘visited’ for each conformer were counted.

Finally, the relative probability of model n was evaluated as P_n_ = N_n_/N_total_, where P_n_ is the population of model n, N_n_ is the total number of conformations visited for model n, and N_total_ is the total steps. The advantages of using MC simulations to estimate conformer probability lie in their good numerical stability and the control that they allow of transition probabilities among several conformers.

## 3. Results and Discussion

Interactions of Aβ dimer with IDE-C or IDE-N domains yield to a distinct IDE conformation.

Three initial IDE-fibril-like Aβ dimer models were constructed: F1, F2, and F3 (Appendix A), and are distinguished by the orientation of the Aβ dimer within the IDE chamber. The specific interactions between the fibril-like Aβ dimer and IDE are detailed in Appendix A. These three simulated models demonstrate the different contacts between the Aβ monomers and IDE domains (Figure 1). While in model F1, most of the contacts are between Aβ monomers and the IDE-C domain, in models F2 and F3, the contacts were mainly with IDE-N. Four different initial IDE-random coil/α-helix Aβ dimer models were constructed: R1, R2, R3, and R4 (Appendix A). The specific interactions between the random coil/α-helix Aβ dimer and IDE are detailed in Appendix A. These four simulated models illustrate different contacts between the Aβ monomers and IDE domains (Figure 2). The number of contacts in models R1, R2, and R3 was larger and less pronounced than in model R4. Moreover, the contacts in models R1, R2, and R3 varied in all domains of IDE, and mainly in the IDE-N domain, but in model R4, the contacts were mainly in the IDE-C domain of IDE.

Interestingly, the dimers that conserved the interactions with IDE-C (models F1 and R4) led to closed-state IDE conformations (Figure 3). All four domains of the closed-state IDE conformations represent low RMSD values (Appendix A). However, the dimers that interacted with IDE-N (models F2, F3, R1, R2 and R3) yielded distinct open-state IDE conformations (Figure 3). The RMSD values of domain D2 of IDE were relatively larger than the other domains, excluding model R3 (Appendix A). It is proposed that the polymorphic Aβ dimer that is entrapped by IDE and contacts with IDE-N agitates domain D2 in IDE and yields an open-state IDE conformation. Moreover, random coil/α-helix Aβ dimers that are more stable in solution and entrapped into IDE lead to relatively low RMSD values of the IDE domains, in comparison to less stable Aβ random coil/α-helix dimers that are entrapped into IDE.

### 3.1. IDE Does Not Inhibit Contacts between Monomers in Fibril-Like Aβ Dimers

The focus of this work was to examine the effect of IDE on the entrapped fibril-like Aβ dimer. Obviously, the IDE conformation influences the dimer. The closed-state IDE conformation that is presented in model F1 yielded relatively low RMSD values of the Aβ dimer, while the open-state IDE conformations that are presented in models F2 and F3, led to relatively high RMSD values of the Aβ dimer (Appendix A). Similarly, the RMSF values along the sequence of Aβ were lower in the case Aβ is entrapped in a closed-state IDE conformation than in the open-state IDE conformation (Appendix A).

The conformation of IDE also affects the contacts between Aβ monomers within the dimer. In model F1, the closed-state IDE blocks up the Aβ dimer, thus strong contacts between Aβ monomers within the dimer are conserved in all primary nucleation domains of aggregation. However, in the open-state IDE conformations (models F2 and F3), one of the domains that play a role in the primary nucleation exhibited less contacts (Figure 4 and Appendix A). The fibril-like Aβ dimer in solution in the absence of IDE presents further contacts between Aβ monomers that lead to a stable “Greek-key parallel sandwich-like” structure [35]. Hence, the entrapment of Aβ by IDE prevents the formation of the Greek-key structure. However, IDE does not inhibit the primary nucleation domain that plays a role in Aβ aggregation: ^17^LVF^20^F sequence. This domain exhibited β-strands in all three cases by which the Aβ dimer is entrapped into the IDE chamber (Appendix A).

### 3.2. IDE Inhibitory Activity Depends on the Stability of the Random Coil/α-Helix Aβ Dimer

It is crucial to investigate the formation of the early-stage Aβ oligomers. Previously, we explored polymorphic Aβ dimers in solution [35], and in membrane [50]. Herein, we applied two polymorphic random coil/α-helix Aβ dimers. The description of the chosen dimers is detailed in the methods section. One Aβ dimer was entrapped into IDE in two orientations, producing two models: R1 and R2. The second Aβ dimer was also entrapped into IDE in two orientations, producing two models: R3 and R4. The second Aβ dimer was energetically more stable than the first Aβ dimer.

While the IDE yielded a closed conformation in model R4, in the other three models, IDE formed an open-state conformation. The closed-state conformation blocked up the dimer and consequently generated low RMSD values of the dimer compared to the other dimers (Appendix A). Respectively, the RMSF values of the dimer in model R4 were lower than the other dimers (Appendix A).

Interestingly, the stable Aβ dimer that was entrapped in IDE (i.e., in models R3 and R4) demonstrated relatively more contacts between the monomers that play a role in primary nucleation, than the less stable Aβ dimer that was entrapped in IDE (i.e., models R1 and R2 (Figure 4)). In model R4, the number of contacts between the monomers within the dimer were the highest among all models because the closed-state IDE conformation blocked up the dimer and promoted primary nucleation.

The contacts between the monomers of each of the two polymorphic dimers were affected by the environment. One of the dimers, which was relatively less stable than the second dimer in solution, exhibited less interactions between the monomers when it was entrapped by IDE (Appendix A), whereas the second dimer that was entrapped into the IDE chamber conserved the interactions between the monomers that were exhibited in solution (Appendix A). Moreover, the closed-state IDE conformation that blocked up the second dimer (model R4) mostly conserved the interactions between the monomers (Appendix A). Finally, the monomers within the dimers that were entrapped in open-state IDE conformations (models R1, R2 and R3) conserved the locations of α-helices along the sequence (Appendix A). However, in the dimer that was entrapped in the closed-state IDE conformation (model R4), the locations of the α-helices extended along the sequence (Appendix A).

### 3.3. Distinct Effects of IDE on the Polymorphic Aβ Dimers

Three polymorphic Aβ dimers were investigated in the current study and were entrapped into the IDE chamber: Aβ fibril-like dimer (producing three IDE-Aβ dimer models: F1–F3), and two Aβ random coil/α-helix dimers (producing four IDE-Aβ dimer models: R1–R4). The hydrogen bond interactions between IDE and the monomers within the Aβ fibril-like dimer that IDE entrapped (in models F1–F3) were relatively larger than in the Aβ random coil/α-helix dimers (models R1–R4) (Appendix A). Moreover, the stable random coil/α-helix dimer that was entrapped in the IDE chamber (in models R3 and R4) had less hydrogen bond interactions between IDE and the monomers compared to the less stable random coil/α-helix dimer (models R1 and R2).

In addition to these hydrogen bond interactions, there are negatively and positively charged residues inside the IDE chamber.. Thus, the interactions between the IDE and each Aβ dimer in models F1–F3 and R1–R4 were not only via hydrogen bond interactions, but also through electrostatic interactions. Interestingly, in all of these models, the negatively charged residues in IDE interacted with the positively charged residues in each of the Aβ dimers (Appendix A). These results are in accordance with the experimental in vitro observations that proposed that amyloid monomers interact with the charged residues in the IDE inner cavity [16].

The solvated IDE was examined in all seven models (F1–F3 and R1–R4). The solvation of IDE depends directly on its conformation in the case IDE entraps a fibril-like Aβ dimer. The closed-state IDE conformation in model F1 exhibited a relatively low solvation during the time of the simulations compared to the open-state IDE conformations in models F2 and F3 (Appendix A). The solvation of IDE in the case IDE entraps distinct random coil/α-helix Aβ dimers were similar, excluding model R2, which demonstrated more solvation of the IDE due to the loop of the IDE that was relatively more solvated than the other models.

The solvation of the monomers within a fibril-like Aβ dimer were relatively larger than the random coil/α-helix Aβ dimers (Appendix A). The solvation of Aβ dimers does not depend on the conformation of IDE, but on the structure of the Aβ dimer. The fibril-like dimer was more extended than the random coil/α-helix Aβ dimers, thus more solvated. The solvation analyses of these dimers provide an interpretation to the conformational solvation energies of these models. The random coil/α-helix Aβ dimers that were less solvated demonstrated lower conformational energies and more populations (Figure 5). Interestingly, the conformational energies of models R3 and R4, by which the most stable random coil/α-helix Aβ dimer was entrapped into the IDE chamber, were lower than those in models R1 and R2, by which the random coil/α-helix Aβ dimer was less stable. Finally, model F1 is slightly more stable and populated than models F2 and F3. Interestingly, in model F1, the structure of IDE was a closed-state conformation, while in models F2 and F3, the structure of IDE was an open-state conformation. It has been proposed that amyloid monomers that contain β-strands that bind to the catalytic chamber, which is in the IDE-N domain, are poorly populated [51]. Interestingly, our results prove this phenomenon. In accordance with this, models F2 and F3, which exhibited interactions of the Aβ dimer with the IDE-N domain demonstrated less population compared to model F1 that comprised the Aβ dimer that binds to IDE-C. It is thus proposed that the IDE conformation affects the conformational energy and the populations in the case a fibril-like Aβ dimer is entrapped in the IDE chamber.

## 4. Conclusions

Extensive studies have reported that IDE entraps and degrades amyloid monomers such as insulin, amylin, and Aβ [13,14,15,16]. The mechanisms by which IDE degrades these amyloid monomers were comprehensively proposed by in vitro and in vivo studies [13,14,15,16]. To date, there is a lack of knowledge on how IDE influences the primary nucleation of Aβ oligomers by the entrapment of early-stage oligomers. This is the first study that illustrates the effect of IDE on entrapped polymorphic Aβ dimers.

Our findings led to several molecular mechanisms by which IDE affects the stability of the polymorphic Aβ dimers and functions to inhibit the formation of early-stage oligomers. Two Aβ monomers may interact to form a fibril-like Aβ dimer or random coil/α-helix Aβ dimer. Obviously, the molecular mechanisms depend on the polymorphic Aβ dimers, but also on the initial contacts between the IDE and Aβ monomers of the dimer (Figure 6).

When IDE entraps the fibril-like Aβ dimer and the monomers are in contact with the IDE-C domain, the IDE blocks up the dimer and stabilizes the contacts between the monomers that play a role in primary nucleation. In this case, IDE does not impede primary nucleation, but plays a role as a “dead-end” molecular chaperone protein, without inhibiting or preventing the contacts between the monomers. Thus, the IDE retains the contacts between the monomers, but prevents the dimer from producing larger oligomers in solution. This result strengthens an in vitro study that proposed that the IDE-C domain plays a crucial role in the regulation of IDE activity [52]. Moreover, our proposed mechanism supports the suggestion by which IDE plays a key role as a chaperone protein in the prevention of amyloid aggregation [26]. However, when IDE entraps the fibril-like Aβ dimer and the monomers are in contact with the IDE-N (or mainly with IDE-N), the open-state IDE conformation slightly impedes the contacts between the monomers.

The inhibitory mechanisms of random coil/α-helix Aβ aggregation depend on the stability of the dimer. When IDE entraps a stable random coil/α-helix Aβ dimer, IDE cannot be capable of preventing the contacts between the monomers. Nevertheless, IDE contributes to the inhibition of Aβ aggregation of these stable dimers by the “dead-end” chaperon activity. When IDE entraps a less stable random coil/α-helix Aβ dimer, IDE can impede the contacts between the monomers. Although IDE not fully encapsulates and blocks up these unstable dimers, it has the quantification to prevent some of the contacts between the monomers that play a role in the primary nucleation of aggregation.

## Figures and Tables

**Figure 1 biomolecules-12-01886-f001:**
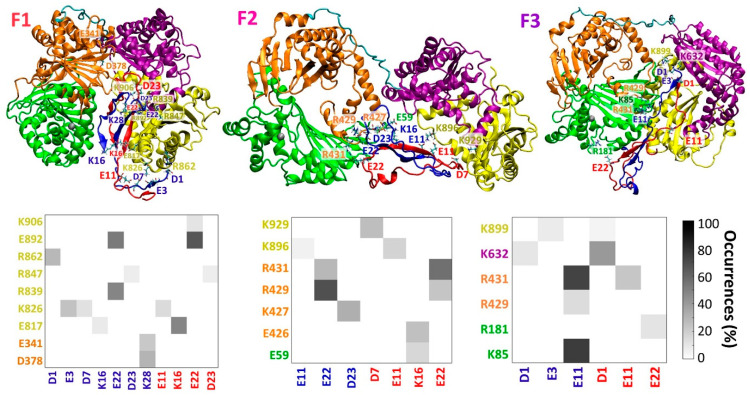
Top: Final simulated structures of IDE-Aβ fibril-like dimer of models F1 (left), F2 (center), and F3 (right). Bottom: Electrostatic interactions contact maps (measured by the occurrences in percentages along the MD simulations) between residues within the IDE domains (domain 1: green; domain 2: orange; domain 3: purple; domain 4: yellow) and Aβ monomers within the dimer for models F1 (left), F2 (center), and F3 (right).

**Figure 2 biomolecules-12-01886-f002:**
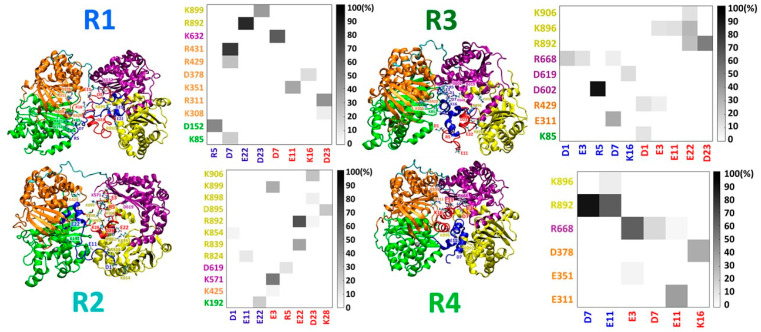
Final simulated structures of IDE-Aβ random coil/α-helix dimer and electrostatic interaction contact maps (measured by the occurrences in percentages along the MD simulations) between residues within the IDE domains (domain 1: green; domain 2: orange; domain 3: purple; domain 4: yellow) and Aβ monomers within the dimer for models R1 (top left), R2 (bottom left), R3 (top right), and R4 (bottom right).

**Figure 3 biomolecules-12-01886-f003:**
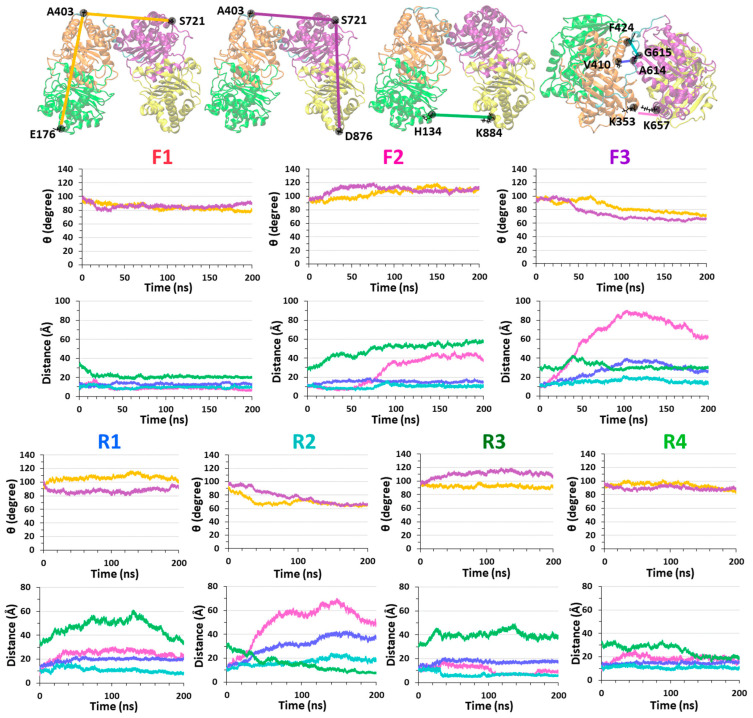
Measured angles [E176-A403-S721 (color: yellow), and A403-S721-D876 (color: purple)] and distances [H134-K884 (color: green), F424-G615 (color: cyan), V410-A614 (color: blue), and K353-K657 (color: pink)] within IDE that estimate the conformation of IDE (open-state/closed-state) along the MD simulations for models F1–F3 and R1–R4.

**Figure 4 biomolecules-12-01886-f004:**
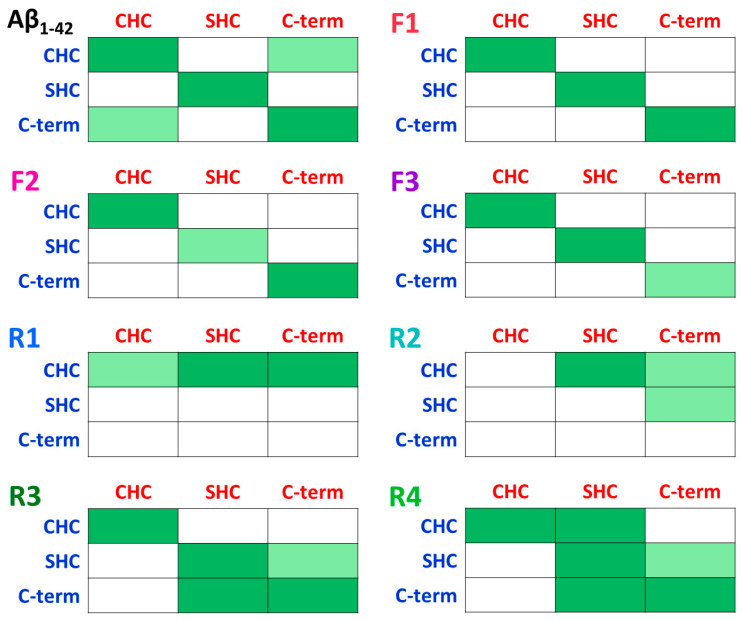
Contacts between the domains (that play a role in primary nucleation) of the monomers (monomer 1: blue; monomer 2: red) within the Aβ dimers in the fibril-like Aβ dimer in solution, in the absence of IDE (top left) [35], and in all models that were studied in the current work: models F1–F3 and R1–R4. Strong interactions (dark green), weak interactions (light green), and none interactions (white).

**Figure 5 biomolecules-12-01886-f005:**
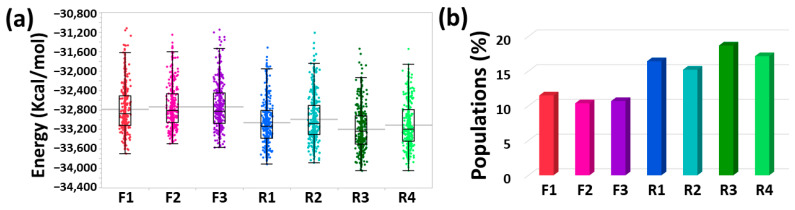
(**a**) Conformational energies computed from the GBMV analysis for all models of the IDE-Aβ dimers: F1 (color: red), F2 (color: pink), F3 (color: purple), R1 (color: blue), R2 (color: cyan), R3 (color: dark green), R4 (color: light green). (**b**) Populations of all IDE-Aβ dimer models.

**Figure 6 biomolecules-12-01886-f006:**
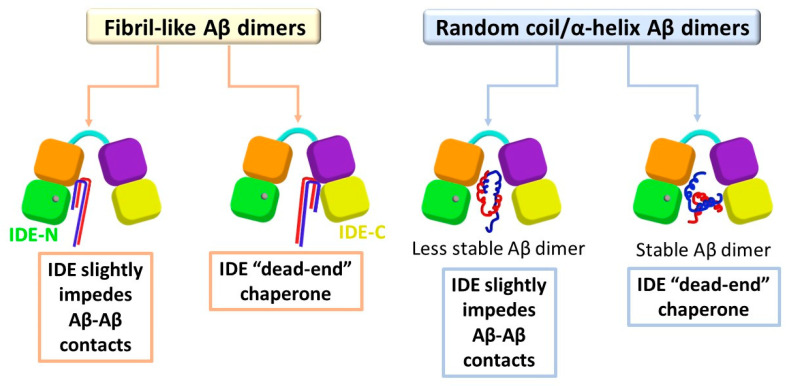
Proposed molecular mechanisms of the IDE effects on polymorphic Aβ dimers. The non-proteolytic inhibitory mechanism slightly impedes the contacts between the monomers. The “dead-end” chaperone activity produces an encapsulated Aβ dimer inside the IDE chamber.

## Data Availability

Not applicable.

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
