# Peer review of "Insights into Non-Proteolytic Inhibitory Mechanisms of Polymorphic Early-Stage Amyloid β Oligomers by Insulin Degrading Enzyme"

_biomolecules, 2022, doi:10.3390/biom12121886_

Round 1

Reviewer 1 Report

The manuscript “Insights into non-proteolytic inhibitory mechanisms of poly-morphic early-stage amyloid β oligomers by insulin degrading enzyme “ by Karina Abramov-Harpaz and Yifat Miller addresses very important question regarding the mechanisms of intermolecular interaction and amyloid inhibiting activity of insulin degrading enzyme (IDE) on Abeta peptide dimers by using molecular dynamic simulation and structural analysis. That is a very challenging task given the size and complexity of active IDE dimer and polymorphism of Abeta dimer. The author minimized the task to investigating two types of Abeta dimers – random coil/alpha-helical and amyloid-like dimers, respectively. The authors have demonstrated the multiple interactions and complex character of inhibitory activity of IDE on Abeta dimer stability, self-assembly and as whether it can serve a role of nucleus in the amyloid formation. When IDE entraps random coil/alpha-helix Aβ dimer, it does not inhibit monomer-monomer interactions, prevents dimer further interactions and serves as the “dead-end” chaperon. When IDE stabilises Abeta fibril-like dimer, it stabilises the contacts between monomers, which plays role in the primary nucleation, but prevents their further self-assembly, again acting as the dead-end chaperon. The research demonstrates the power of MD in the intricate analysis of molecular interactions and sheds light on the molecular interactions underlying the functional role of IDE in preventing amyloid development. The research is very clearly and comprehensively presented. I recommend it for publication without any reservations. 

Author Response

We would like to thank the reviewer for the positive comments.

Reviewer 2 Report

The authors should address the following comments:

1-Is there another type of involved interaction other than those discussed in the manuscript?

2-Are the criteria for detecting the kind of interaction sufficient?

3-The environment of the considered system is provided by the water molecules, but for an in vitro amyloid fibrillation study, it is necessary to add some salts such NaCl, KCl,..So, why the authors did not consider this issue?

4-Is it possible to control pH value in the MD simulation as the authors aimed to compare with the results of in vitro studies?   

Author Response

The authors should address the following comments:

1-Is there another type of involved interaction other than those discussed in the manuscript?

Response:

We investigated all types of interactions. The specific interactions that we investigated are between the two Aβ monomers that play role in the primary nucleation of Aβ aggregation, which are mainly hydrophobic interactions. In addition, the interactions between each Aβ monomer and IDE were investigated. Since the IDE chamber contains mainly positively and negatively charged residues, the interactions between IDE and each Aβ monomer are electrostatic interactions. Finally, all hydrogen bonds interactions were also computed. 

2-Are the criteria for detecting the kind of interaction sufficient?

Response:

The criteria for detecting the interactions were based on two references - Refs 46 and 47 (Kumar, S.; Nussinov, R., Close-range electrostatic interactions in proteins. Chembiochem 2002, 3, 604-17; Deloof, H.; Nilsson, L.; Rigler, R., Molecular-Dynamics Simulation of Galanin in Aqueous and Nonaqueous Solution. Journal of the American Chemical Society 1992, 114, 4028-4035.) These criteria were proved as sufficient and are based on experimental observations.

3-The environment of the considered system is provided by the water molecules, but for an in vitro amyloid fibrillation study, it is necessary to add some salts such NaCl, KCl,..So, why the authors did not consider this issue?

Response:

We considered counter ions, which is NaCl as noted in page 6 of the revised manuscript. We now included the type of the salt: NaCl.

4-Is it possible to control pH value in the MD simulation as the authors aimed to compare with the results of in vitro studies?

Response:

It is possible to control pH value in the MD simulations. Previously, we applied MD simulations in various pH values: 3-4, 5-6 and 7. The determination/estimation of the pH conditions were based on the following: To study the effect of the pH on the structural stability and the populations, we used different protonation states of titratable side chains to simulate constructed models at different pH values. For pH = 7, only the positive charged residues (Lys and Arg) were protonated. For pH = 5–6, all histidines were also protonated, whereas for pH = 3–4, all Lys, Arg, His, Glu, and Asp residues were protonated.